# Survival Study: International Multicentric Minimally Invasive Liver Resection for Colorectal Liver Metastases (SIMMILR-2)

**DOI:** 10.3390/cancers14174190

**Published:** 2022-08-29

**Authors:** Andrew A. Gumbs, Roland Croner, Eric Lorenz, Andrea Benedetti Cacciaguerra, Tzu-Jung Tsai, Lee Starker, Joe Flanagan, Ng Jing Yu, Elie Chouillard, Mohammad Abu Hilal

**Affiliations:** 1Departement de Chirurgie Digestive, Centre Hospitalier Intercommunal de Poissy/Saint-Germain-en-Laye 10, Rue du Champ Gaillard, 78300 Poissy, France; 2Department of General-, Visceral-, Vascular- and Transplantation Surgery, University of Magdeburg, Haus 60a, Leipziger Str. 44, 39120 Magdeburg, Germany; 3Department of Surgery, Koo Foundation Sun Yat-Sen Cancer Centre, Taipei 112, Taiwan; 4Department of Surgical Oncology, Morristown Medical Center, Morristown, NJ 07960, USA; 5Unità Chirurgia Epatobiliopancreatica, Robotica e Mininvasiva, Fondazione Poliambulanza Istituto Ospedaliero, Via Bissolati, 57, 25124 Brescia, Italy

**Keywords:** epatectomy, laparoscopic, robotic, minimally invasive, liver resection, colorectal liver metastasis, multicentric, international, laparoscopy, robot-assisted, overall survival, recurrence free survival

## Abstract

**Simple Summary:**

This is a follow-up study of an international multicentric cohort after minimally invasive liver resection (SIMMILR-1) from five international centers evaluating long-term outcomes after minimally invasive liver resection for patients with three or fewer colorectal liver metastases that measure less than or equal to 3 cm, or a solitary tumor less than or equal to 5 cm (Milan Criteria). Propensity score matching was done to reduce bias. Comparisons were done between open, laparoscopic and robotic liver resections. Laparoscopic and robotic approaches may have short-term benefits when compared to open hepatectomy (SIMMILR-1), but no long-term benefits on survival have been identified as of yet (SIMMILR-2). Larger trials that are randomized and controlled are needed to better ascertain whether or not one of the surgical approaches has any long-term advantages or disadvantages.

**Abstract:**

Introduction: Study: International Multicentric Minimally Invasive Liver Resection for Colorectal Liver Metastases (SIMMILR-CRLM) was a propensity score matched (PSM) study that reported short-term outcomes of patients with CRLM who met the Milan criteria and underwent either open (OLR), laparoscopic (LLR) or robotic liver resection (RLR). This study, designated as SIMMILR-2, reports the long-term outcomes from that initial study, now referred to as SIMMILR-1. Methods: Data regarding neoadjuvant chemotherapeutic (NC) and neoadjuvant biological (NB) treatments received were collected, and Kaplan–Meier curves reporting the 5-year overall (OS) and recurrence-free survival (RFS) for OLR, LLR and RLR were created for patients who presented with synchronous lesions only, as there was insufficient follow-up for patients with metachronous lesions. Results: A total of 73% of patients received NC and 38% received NB in the OLR group compared to 70% and 28% in the LLR group, respectively (*p* = 0.5 and *p* = 0.08). A total of 82% of patients received NC and 40% received NB in the OLR group compared to 86% and 32% in the RLR group, respectively (*p* > 0.05). A total of 71% of patients received NC and 53% received NB in the LLR group compared to 71% and 47% in the RLR group, respectively (*p* > 0.05). OS at 5 years was 34.8% after OLR compared to 37.1% after LLR (*p* = 0.4), 34.3% after OLR compared to 46.9% after RLR (*p* = 0.4) and 30.3% after LLR compared to 46.9% after RLR (*p* = 0.9). RFS at 5 years was 12.1% after OLR compared to 20.7% after LLR (*p* = 0.6), 33.3% after OLR compared to 26.3% after RLR (*p* = 0.6) and 22.7% after LLR compared to 34.6% after RLR (*p* = 0.6). Conclusions: When comparing OLR, LLR and RLR, the OS and RFS were all similar after utilization of the Milan criteria and PSM. Biological agents tended to be utilized more in the OLR group when compared to the LLR group, suggesting that highly aggressive tumors are still managed through an open approach.

## 1. Introduction

Study: International Multicentered Minimally Invasive Liver Resection (SIMMILR) was a study focused on colorectal liver metastases (CRLMs) from six international centers where, after propensity score matching (PSM), data from only five centers remained [1]. In an effort to make as fair a comparison as possible, only CRLMs that adhered to the Milan criteria, a classification system traditionally reserved for hepatocellular carcinoma [2] but increasingly used for CRLM, were studied [3]. Potential bias between liver resections performed through an open approach and minimally invasive techniques was reduced by controlling for preoperative factors. The initial study (SIMMILR-CRLM), hereafter designated as SIMMILR-1, focused on the analysis of short-term outcomes and compared open liver resection (OLR), laparoscopic (LLR) and robotic-assisted liver resection (RLR) to each other [1]. After PSM in SIMMILR-1, blood loss, hospitalization and morbidity rates were statistically significantly decreased in the LLR group when compared to the OLR group. RLR had significantly less estimated blood loss when compared to both OLR and LLR.

This study, SIMMILR-2, aimed to assess whether any differences can be found in a comparison of these three surgical approaches in terms of long-term outcomes, namely, overall survival (OS) and recurrence-free survival (RFS). Data exist comparing OLR and LLR but are limited to short-term outcomes, with oncological data limited to showing the equivalency of resection margin status [4]; similarly, short-term outcomes [5,6] and margin status have been compared between OLR and RLR for CRLM, with similar results seen in both groups [7,8]. Due to the great disparity in how studies and data are reported, with synchronous and metachronous lesions often being reported together, we only included data from patients who presented with synchronous lesions, and we controlled for neoadjuvant chemotherapeutic and biological treatments received.

## 2. Methods

Laparoscopic and robotic-assisted liver resections were both considered minimally invasive. Previously used inclusion and exclusion criteria were reported in SIMMILR-1. Resections were divided by the approach used: open liver resection (OLR), laparoscopic liver resection (LLR) and robotic liver resection (RLR). Patients with tumors that were not within the Milan criteria were excluded. The defined Milan criteria were patients with 2–3 secondary tumors measuring ≤3 cm or a solitary metastasis ≤5 cm. Preoperative factors that were controlled during the PSM included, but were not limited to, the number of lesions, size of lesions, location in the deep hepatic segments and the percentage of patients who received neoadjuvant treatments. Neoadjuvant treatments were given when patients did not meet the Milan criteria or had enlarged lymph nodes or lesions next to large hepatic vessels when tumor shrinkage could result in a smaller hepatectomy and when upfront liver resection was otherwise not possible according to the local multidisciplinary tumor board [9,10].

Any remaining heterogeneity found was further analyzed after propensity score matching (PSM) to decrease any potential bias. Specifically, we looked for any differences in the number of metachronous vs. synchronous lesions, the percentage of patients who received neoadjuvant chemotherapy (NC), the type of chemotherapy used, the percentage of patients who received neoadjuvant biologicals (NBs) and the type administered. The biological antibody Bevacizumab was added for tumor patients with an elevated preoperative serum CEA level > 200 ng/mL or more than 1 CRM; alternatively, the antibody Cetuximab was added for KRAS exon 2 wild-type tumors that met these criteria [11].

For the survival analyses, only synchronous lesions were used to ensure adequate numbers. Due to the heterogeneity of adjuvant chemotherapeutic treatments, these data were not included. Additionally, any patients who underwent liver resection combined with radiofrequency or microwave ablation were excluded [12]. Although the anterior approach was used during minimally invasive major hepatectomy, both anterior and posterior approaches were used during open procedures [13]. Although the safety of concomitant liver resection and colorectal resection has even been shown in robotic-assisted patients, those who underwent simultaneous colorectal surgery were not included in any of the 3 groups [14].

Statistical data analysis was performed using the Social Science Statistics software (www.socscistatistics.com, accessed on 1 July 2022) and SPSS (version 26; IBM, Armonk, NY, USA). Prism 8: GraphPad software (https://www.graphpad.com/scientific-software/prism/, accessed on 1 July 2022) was used to generate Kaplan–Meier curves. Categorical data (nominal/ordinal) are presented as absolute (n) and/or relative values (%). Differences between the groups were tested using Pearson’s χ^2^ or Student’s *t*-test (if at least one cell had a cell count of less than 5). Continuous data were expressed as mean (SD—standard deviation). Differences between continuous variables were analyzed using the Mann–Whitney U-test for continuous variables with <200 distinct values. For all analyses, differences with a two-sided *p*-value < 0.05 were considered to be significant (no adjustment for multiplicity).

## 3. Results

After the initial PSM, no differences were found in terms of the percentage of synchronous vs. metachronous lesions, patients undergoing neoadjuvant chemotherapy or patients receiving neoadjuvant biologicals among the three approaches (Table 1). When OLR was compared to LLR, a slightly higher percentage of synchronous CRLM was found in the LLR group when compared to the OLR group, but this finding failed to reach statistical significance (*p* = 0.06). The percentage of patients who received neoadjuvant chemotherapy was comparable between both groups; however, although not statistically significant, those who underwent OLR tended to receive biological agents more frequently in the neoadjuvant setting (*p* = 0.08). When OLR was compared with RLR and LLR was compared with RLR, no differences even tended to approach significance.

When all OLR patients from PSM were compared to all of the patients who underwent minimally invasive liver resection (MILR), LLR and RLR, significantly more synchronous lesions were found in the MILR group when compared to the OLR group, 80% vs. 70%, respectively (*p* = 0.02) (Table 2). Furthermore, fewer biological agents tended to be used in the MILR group (30%) when compared to the OLR group (38%), but this only approached statistical significance (*p* = 0.08).

Because of the similarity in oncological data, survival curves were developed but limited to patients with synchronous lesions. Follow-up data available to create meaningful Kaplan–Meier Curves of patients with metachronous CRLMs were not sufficient. Overall survival (OS) and recurrence-free survival (RFS) were not statistically different regardless of the operative approach. Survival curves were not generated for the data in Table 2 because this comparison no longer followed PSM.

After PSM, overall survival (OS) at 1, 3 and 5 years was 91.9%, 60.0% and 34.8% with a median survival of 45 months after OLR, compared to 87.1%, 54.0% and 37.1% with a median survival of 39 months after LLR, respectively (*p* = 0.435) (Figure 1A). When OLR and RLR were compared, OS at 1, 3 and 5 years was 93.3%, 53.3% and 34.3% with a median survival of 37 months after OLR, compared to 100%, 75.0% and 46.9% with a median survival of 46 months after RLR, respectively (*p* = 0.433) (Figure 2A). Similarly, OS at 1, 3 and 5 years was 94.4%, 68.1% and 30.3% with a median survival of 53 months after LLR, compared to 100%, 75% and 46.9% after RLR with a median survival of 46 months, respectively (*p* = 0.908) (Figure 3A).

Recurrence-free survival (RFS) at 1, 3 and 5 years was 61.0%, 30.5% and 12.1% with a median survival of 21 months after OLR, compared to 64.2%, 33.7% and 20.7% with a median survival of 20 months after LLR, respectively (*p* = 0.645) (Figure 1B). RFS at 1, 3 and 5 years was 46.7%, 33.3% and 33.3% with a median follow-up of 12 months after OLR, compared to 78.8%, 26.3% and 26.3% with a median follow-up of 22 months after RLR, respectively (*p* = 0.604) (Figure 2B). RFS at 1, 3 and 5 years was 73.7%, 45.3% and 22.7% after LLR with a median follow-up of 28 months, compared to 80.8%, 34.6% and 34.6% with a median follow-up of 22 months after RLR, respectively (*p* = 0.606) (Figure 3B).

## 4. Discussion

After limiting patients to those who met the Milan criteria and had synchronous CRLMs and after PSM, the overall and recurrence-free survival were similar when all three surgical approaches were compared (Figure 1, Figure 2 and Figure 3). Nonetheless, minimally invasive approaches have many short-term benefits that would argue for the utilization of a minimally invasive approach when patients with synchronous CRLMs that meet the Milan criteria become surgical candidates. An early criticism of minimally invasive liver resection was that more major resections were performed when compared to OLR [15]. This was accounted for by also using the degree of resection as one of the criteria to be controlled for in the initial study, SIMMILR-1. The Milan criteria were used to create a more useful approach to the management of patients with CRLM.

Another potential benefit of using the Milan criteria for patients with CRLM is the avoidance of patients with very early recurrences, which can reduce the 5-year OS from 44.5% to 17.3% [16]. Although we do not mean to imply that patients with more than three CRLMs should not be considered for surgery, we do acknowledge that patients have decreased survival the more metastases they have [17]. Notably, some studies have found OS rates of 30% in patients with 10 or more metastases, as long as their resection margins were all R0 [18]. Aside from the largest diameter and the number of CRLMs, the number of positive lymph nodes on preoperative imaging should also be considered, because patients with six or more positive lymph nodes do as well as patients with non-resectable metastatic disease [19]. Although patients with CRLM who also have pulmonary metastasectomy can have 5-year OS and RFS rates of 36% and 10%, respectively, these patients were excluded from our study [20].

Unfortunately, the various centers in SIMMILR-2 did not routinely use the IWATE scoring system to define the lesions preoperatively. We attempted to overcome this by identifying the percentage of lesions in the deep segments, but admittedly, future studies will need to record the IWATE score preoperatively to permit a more accurate comparison between the surgical approaches. Interestingly, the location of CRLM in the left or right liver may also have to be taken into account. A study found that lesions in the right liver were associated with a worse prognosis in the first 10 months, but that with time, this difference inverted as the survival curves crossed with right-sided lesions, conferring a survival benefit later on [21]. Conversely, lesions in the deep hepatic segments seem to confer worse survival, with patients undergoing concomitant ablations also suffering a worse prognosis [22].

There was a tendency for more lesions to be synchronous in the LLR group when compared to the OLR group. Although this did not attain statistical significance, this potential bias was hopefully reduced by only including synchronous lesions in the survival curves. When evaluating the survival curves, it is important to remember that synchronous CRLMs will have decreased survival when compared to metachronous lesions. As a result, our survival may seem unfairly decreased in the minimally invasive groups, particularly the LLR group [23,24,25,26,27]. Additionally, although there was a tendency for the Pringle maneuver to be more commonly used after OLR, this was not statistically significant, and furthermore, studies have shown that utilization of this technique does not seem to influence survival [28]. Although resectability was determined locally by a multidisciplinary oncology tumor board, an interesting study from Finland found that centralized resectability assessment may enable improved resectability and survival rates [29]. Future randomized controlled trials should perhaps include a centralized resectability assessment to truly obtain consistent and useful data.

In the literature, 5-year OS and DFS traditionally range between 22–51.6% and 13.6–31.9%, respectively. However, number of metastases >3 is also dependent on the timing of presentation (synchronous vs. metachronous) and may cause increased postoperative complication rates, such as infection. Notably, infection might have more of an influence on survival than the severity of complications [30,31,32,33,34,35,36,37]. When OLR is compared to LLR for CRLM, 5-yr OS has been found to not be significantly different, with 63% after OLR and 54% after LLR, with a 5-yr RFS of 38% and 36%, respectively [38]. When only synchronous CRLMs are analyzed, the reported 5-year OS decreases to 25.5–41.1% with a 5-yr RFS of 16.8% [39,40], and 3-year survival ranges from 47–58%, which is consistent with our results [41]. Up to one-third of patients alive at 5 years will ultimately succumb to recurrent disease; because of this, 10-year survival may be a better indicator of cure in patients with CRLM [28]. Despite the fact that some centers recommend surveillance for only 3 years in patients with node-negative disease and a disease-free interval less than 12 months between resection of the primary and CRLM, we still like to follow our patients with CRLM for 10 years [42]. The first randomized controlled trial comparing OLR to LLR for the management of CRLMs (OSLO-COMET) combined patients with synchronous and metachronous lesions, with only 61.9% and 56.4% of synchronous lesions in the OLR and LLR groups, respectively [43]. This probably explains the superior 5-year OS survival seen in their open and laparoscopic groups (55% and 54%, respectively) when compared to our results [43].

Although repeat hepatectomy is feasible even after open major hepatectomy, it may occur more often if the initial liver resection is performed minimally invasively. The first study to analyze the feasibility of repeat laparoscopic hepatectomy came from the group at the Institut Mutualiste Montsouris in Paris [44]. Patients with recurrent CRLMs who are able to undergo liver resection have superior 5-year OS at 51–71.8% compared to 19–45% for patients who are unable to undergo repeat hepatectomy [45,46,47]. In a study comparing open vs. laparoscopic two-stage liver resection, only patients who initially underwent laparoscopic liver resection ultimately underwent a liver resection if they developed a recurrence in the liver (56% vs. 0%, *p* = 0.006) [48]. When OLR and LLR resections were compared in another study, short-term outcomes were similar for re-resection of CRLM; however, the LLR group had a higher incidence of liver insufficiency [49].

Although simultaneous colorectal and liver resections can be performed for CRLM with a similar 3-year OS, regardless of whether resections were simultaneous (66.1%) or staged (62.3%), simultaneous procedures were excluded from our PSM [50]. This referenced study is also notable because, like SIMMILR-2, it is one of the few survival studies analyzing liver resection for CRLM that exclude metachronous lesions, and it reports 3-year survival rates that are similar. Regardless of whether or not a liver-first strategy or a traditional approach with colorectal resection followed by a second-stage liver resection is utilized, approximately 35% of patients will not undergo the intended liver and/or colorectal resection. Because of this, when possible, simultaneous resection of both the primary and secondary lesions should be considered [51].

Even though some studies have shown no benefit in terms of survival in patients with CRLM who receive neoadjuvant chemotherapy, our preference was to administer it to all patients unless they had a resectable solitary CRLM, no enlarged lymph nodes and no evidence of extra-hepatic disease [10]. Due to the fact that margin status is an important determinant of survival after liver resection for CRLM, every effort has traditionally been made to ensure a complete resection [9,52]. However, a recent study from Switzerland indicates that R1 status does not impact local recurrence rates or even overall survival; as a result, indications for liver resection could perhaps be broadened [53]. A study from Japan found that only 15% of patients with R1 resections recurred at the resection margin, with the majority recurring elsewhere in the liver or beyond [54]. This observation is contradicted by an Italian study that found decreased survival after R1 resections [55]. Even though studies have shown that patients with CRLM who fail first-line neoadjuvant chemotherapy but respond to second-line treatment and ultimately undergo liver resection have similar survival, we excluded these patients in an effort to minimize bias as much as possible [24,56].

One important determinant of long-term survival is the ability to start and tolerate adjuvant chemotherapy [57,58]. Future studies will analyze whether or not patients were able to start adjuvant treatments sooner or later depending on which approach was used for liver resection. The lack of data on adjuvant chemotherapy in this trial is a limitation, but as mentioned, the heterogeneity in treatments given precludes any meaningful comments and again highlights the need for large randomized controlled trials. This is particularly true because survival has been shown to have improved with newer chemotherapeutic agents, with one study showing an improvement in 5-year OS from 33.3% to 49.0% since 2005 due to changes in modern adjuvant treatments [59].

## 5. Future Directions

Although advances in AI will enable enhanced computer vision and more autonomous actions in the operating room, it does not appear that the type of approach used has a significant impact on survival for patients with synchronous CRLMs that adhere to the Milan criteria. In addition to improvements in chemotherapeutic and biological agents, increased utilization of big data analytic tools in determining which patients are good candidates for surgery and what types of neoadjuvant and adjuvant treatments to administer may help improve OS and RFS even further [4,12,55]. More complex models may be used in the future to better predict survival. One such example is the contour prognostic model that uses the size of the largest hepatic metastasis, the total number of metastases and RAS mutation status [3]. Although not widely used, except in large referral centers, it is hoped that a broader implementation of artificial intelligence (AI) in the future will make the routine use of models such as these more prevalent [57].

Newer collaborative robots that are handheld or do not use a console may also become useful in the surgeon’s armamentarium against CRLM and may actually permit a safer adoption of more automatic and autonomous actions in the operating room [58,59,60,61]. Modern-day liver surgeons will need expertise in open, laparoscopic and robotic-assisted techniques to offer the best personalized surgical care to their patients with CRLM. An analysis of long-term outcomes may show that the minimally invasive approach has advantages for the surgical management of CRLM once more surgeons are able to receive the proper mentorship and experience in these various techniques [62].

## 6. Conclusions

Regardless of whether open, laparoscopic or robotic hepatectomy was performed, no significant differences in OS or RFS were found, and survival rates were similar. However, the short-term benefits observed from minimally invasive liver resection and decreased scar formation may allow adjuvant chemotherapy to be started sooner and to increase re-operations for hepatic recurrences. Randomized controlled trials with longer-term follow-up are still needed, particularly in the robotic arm. Ultimately, advances in chemotherapeutic and biological agents and AI may enable the most interesting improvements in the management of patients with CRLM.

## Figures and Tables

**Figure 1 cancers-14-04190-f001:**
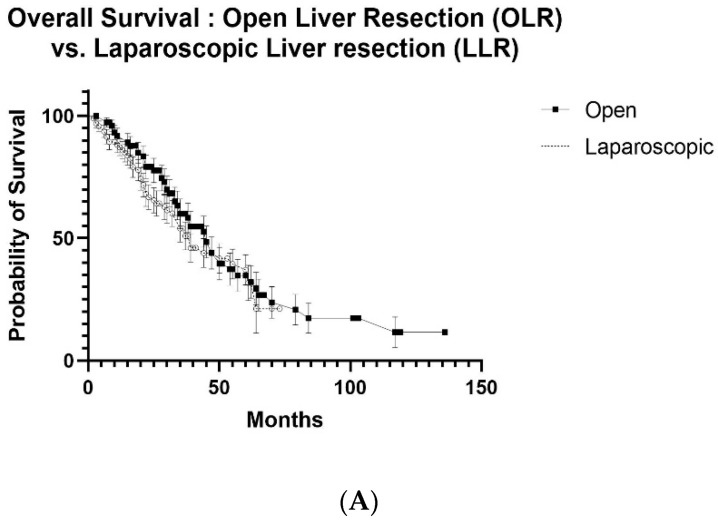
(**A**) Overall survival curves for open liver resection (OLR) vs. laparoscopic liver resection (LLR) after propensity score matching (PSM) for colorectal liver metastases (CRLMs), (**B**) recurrence-free survival curves for open liver resection (OLR) vs. laparoscopic liver resection (LLR) after propensity score matching (PSM) for colorectal liver metastases (CRLM).

**Figure 2 cancers-14-04190-f002:**
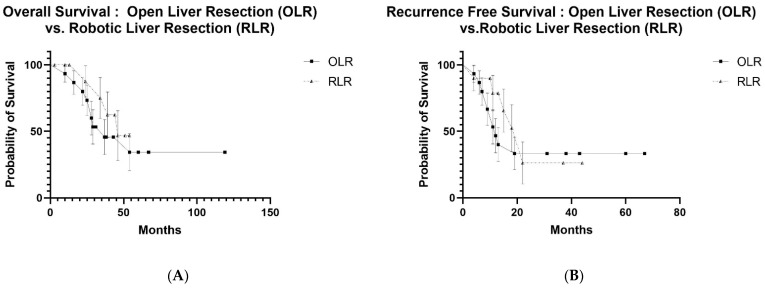
(**A**) Overall survival curves for open liver resection (OLR) vs. robotic liver resection (RLR) after propensity score matching (PSM) for colorectal liver metastases (CRLMs), (**B**) recurrence-free survival curves for open liver resection (OLR) vs. robotic liver resection (RLR) after Propensity Score Matching (PSM) for colorectal liver metastases (CRLMs).

**Figure 3 cancers-14-04190-f003:**
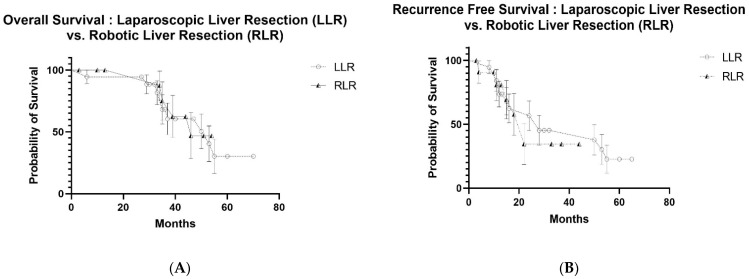
(**A**) Overall survival curves for laparoscopic liver resection (LLR) vs. robotic liver resection (RLR) after propensity score matching (PSM) for colorectal liver metastases (CRLM), (**B**) recurrence-free survival curves for laparoscopic liver resection (LLR) vs. robotic liver resection (RLR) after propensity score matching (PSM) for colorectal liver metastases (CRLMs).

**Table 1 cancers-14-04190-t001:** Oncological factors of open liver resection (OLR) vs. laparoscopic liver resection (LLR), open liver resection (OLR) vs. robotic liver resection (RLR) and laparoscopic liver resection (LLR) vs. robotic liver resection (RLR), including the type of neoadjuvant chemotherapy and biological administered.

	OLR n = 142	LLR n = 142	*p*-Value	OLR n = 22	RLR n = 22	*p*-Value	LLR n = 21	RLR n = 21	*p*-Value
**Synchronous** **(%)**	99 (69.7)	114 (80.3)	*0.055*	15 (68.1)	17 (77.3)	0.735	18 (81.8)	16 (76.2)	0.697
**Neoadjuvant chemotherapy (%)**	104 (73.2)	99 (69.7)	0.5	18 (81.8)	19 (86.4)	1	15 (71.4)	15 (71.4)	1
Folfox (%) Average cycles (Range)	56 (39.4) 6 (4.5–8)	43 (30.3) 5 (4–6)	0.122	10 7 (6–12)	11 6 (3–12)	0.431	8 (53.3) 5 (4–6)	8 (53.3) 6 (3–12)	0.424
Folfirinox (%) Average cycles (Range)	37 (26.1) 8 (6–12)	36 (25.4) 6 (3–9)		6 8 (6–12)	7 6 (3–12)		5 (33.3) 6 (3–9)	6 (40) 7 (3–12)	
5-FU alone (%) Average cycles (Range)	11 (7.7) 6 (3–12)	20 (14.1) 4 (3–7)		2 7 (5–9)	1 6 (6)		2 (13.3) 5 (3–7)	1 (6.7) 6 (6)	
**Neoadjuvant biologicals (%)**	n = 54 (38.0)	n = 39 (27.5)	*0.076*	*9* *(40.1)*	*7* *(31.8)*	*0.755*	*8* *(53.3)*	*7* *(46.7)*	*1*
Bevacizumab (%) Average cycles (Range)	37 (26.1) 6 (5–12)	32 (19.0) 5 (4–7)	0.158	6 (22.7) 6 (5–12)	4 (18.2) 8 (6–12)	1	6 (40) 6 (4–7)	4 (26.7) 6 (5–7)	0.608
Cetuximab (%) Average cycles (Range)	17 (12.0) 6 (5–8)	7 (3.5) 5 (4–6)		3 (13.6) 6 (5–8)	3 (13.6) 6 (6–7)		2 (13.3) 6 (5–7)	3 (20) 6 (5–7)	

**Table 2 cancers-14-04190-t002:** Oncological factors after propensity score matching of all open liver resections (OLRs) compared to all laparoscopic liver resections (LLRs) and robotic liver resections (RLRs) combined, including the type of neoadjuvant chemotherapy and biological administered.

	OLR n = 164	MILR n = 206	*p*-Value
**Synchronous** **(%)**	114 (69.5)	165 (80.1)	0.021
**Neoadjuvant chemotherapy** **(%)**	122 (74.4)	148 (71.8)	0.638
Folfox (%) Average cycles (Range)	66(40.2) 6 (4.5–12)	70 (34.0) 5 (3–12)	0.341
Folfirinox (%) Average cycles (Range)	43 (26.2) 8 (6–12)	54 (26.2) 6 (3–12)	
5-FU alone (%) Average cycles (Range)	13 (7.9) 6 (5–9)	24 (11.7) 4 (3–7)	
**Neoadjuvant biologicals** **(%)**	63 (38.4)	61 (29.6)	*0.078*
Bevacizumab (%) Average cycles (Range)	43 (26.2) 6 (5–12)	46 (22.3) 5 (4–12)	
Cetuximab (%) Average cycles (Range)	20 (12.2) 6 (5–8)	15 (13.6) 6 (4–8)	

## Data Availability

Data available upon reasonable request.

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
