# Peer review of "Survival Study: International Multicentric Minimally Invasive Liver Resection for Colorectal Liver Metastases (SIMMILR-2)"

_cancers, 2022, doi:10.3390/cancers14174190_

Round 1

Reviewer 1 Report

In this paper, the authors did a study on International Multicentric Minimally Invasive Liver Resection for Colorectal Liver Metastases (SIMMILR-CRLM/SIMMILR-1) on short-term outcomes of patients with CRLM who met the Milan Criteria and underwent either open (OLR), laparoscopic (LLR) or robotic liver resection (RLR).

In this article, the patient samples are well selected, however the way the authors phrasing their study as well as the way they are presenting the data is hard to interpret. For instance, in line 27-29, the author said: “A total of 73% of patients received 5-FU based neoadjuvant chemotherapy and 38% neoadjuvant biological agents in the OLR compared to 69.7% and 27.5% in the LLR, respectively (p=0.5 and p=0.08).” It is a long sentence and with a word “received” missing after “38%”. In just the following from line 29 to 31,the author said:” When OLR and RLR were compared the percentages were 81.8%/40.1% and 86.4%/31.8%, respectively (p=0.4 and 0.8). When LLR and RLR were compared the percentages were 71.4%/53.3% and 71.4%/46.7%, respectively (p=0.4 and 1). It takes a long time for readers to track back and understand what the numbers “81.8%/40.1%”, “86.4%/31.8%”,”71.4%/53.3%”and “71.4%/46.7%” are corresponding to.

Also in Table 1-3, authors compared OLR, LLR and RLR separately by pairs, which is really hard for readers to interpret and judge the three approaches OLR, LLR and RLR as a whole. Can authors combine and normalize these three tables into one?

In addition, there are many 2-spaced spaces between the two words, such as 2-space between “LLR” and “compared” in line 33, “collected” and “and” in line 24, and “respectively” and “(p=0.433)” in line 145.

I suggest 1. The authors read through the whole articles over and change      the way they are presenting to the readers;

                2. Make a clearer way of presenting the data in Table1-3 before they resubmit their manuscript.

Author Response

RESPONSE TO REVIEWER 1

In this paper, the authors did a study on International Multicentric Minimally Invasive Liver Resection for Colorectal Liver Metastases (SIMMILR-CRLM/SIMMILR-1) on short-term outcomes of patients with CRLM who met the Milan Criteria and underwent either open (OLR), laparoscopic (LLR) or robotic liver resection (RLR).

In this article, the patient samples are well selected, however the way the authors phrasing their study as well as the way they are presenting the data is hard to interpret. For instance, in line 27-29, the author said: “A total of 73% of patients received 5-FU based neoadjuvant chemotherapy and 38% neoadjuvant biological agents in the OLR compared to 69.7% and 27.5% in the LLR, respectively (p=0.5 and p=0.08).” It is a long sentence and with a word “received” missing after “38%”. 

This was corrected and the word “received” was added.

The ABSTRACT, INTRODUCTION and DISCUSSION were significantly modified to help streamline the transmission of information. In particular, the discussion had over 4 paragraphs removed.

The METHODS and RESULTS also underwent modification, with the majority of changes coming in the new Tables and Figure layouts (see comment below).

In just the following from line 29 to 31,the author said:” When OLR and RLR were compared the percentages were 81.8%/40.1% and 86.4%/31.8%, respectively (p=0.4 and 0.8). When LLR and RLR were compared the percentages were 71.4%/53.3% and 71.4%/46.7%, respectively (p=0.4 and 1). It takes a long time for readers to track back and understand what the numbers “81.8%/40.1%”, “86.4%/31.8%”,”71.4%/53.3%”and “71.4%/46.7%” are corresponding to.

This section was changed and now reads :

Methods: Data regarding neoadjuvant chemotherapeutic (NC) and neoadjuvant biological (NB) treatments received were collected  and Kaplan Meier curves, reporting the 5-year overall (OS) and recurrence free survival (RFS) among OLR, LLR and RLR, were created for patients who presented with synchronous lesions only, as there was not enough follow-up for patients with metachronous lesions. Results: A total of 73% of patients received NC and 38% received NB in the OLR group compared to 70% and 28% in the LLR group, respectively (p=0.5 and p=0.08). A total of 82% of patients received NC and 40% received NB in the OLR group compared to 86% and 32% in the RLR group, respectively (p>0.05). A total of 71% of patients received NC and 53% received NB in the LLR group compared to 71% and 47% in the RLR group, respectively (p>0.05).

Also in Table 1-3, authors compared OLR, LLR and RLR separately by pairs, which is really hard for readers to interpret and judge the three approaches OLR, LLR and RLR as a whole. Can authors combine and normalize these three tables into one?

Table 1-3 were combined into 1 Table, now labeled (Table 1). The whole table was cleaned up and hopefully it is easier to interpret now.

Additionally a new Table that was requested by a reviewer was added.

Table 2. Oncologic factors after Propensity Score Matching of all Open Liver Resections (OLR) compared to all Laparoscopic Liver Resections (LLR) and Robotic Liver Resections (RLR) combined, including type of neoadjuvant chemotherapy and biological administered.

In addition, there are many 2-spaced spaces between the two words, such as 2-space between “LLR” and “compared” in line 33, “collected” and “and” in line 24, and “respectively” and “(p=0.433)” in line 145.

All double spaces were searched for and removed. There were 27 in total, thank you very much for bringing our attention to this.

I suggest 1. The authors read through the whole articles over and change      the way they are presenting to the readers;

                2. Make a clearer way of presenting the data in Table1-3 before they resubmit their manuscript.

Please see above changes to the Tables, also note that the Figures were now consolidated into 3 Figures from 6, upon request from one of the reviewers. Overall Survival and Recurrence Free Survival appear in the same figure and are labeled A and B, respectively.

Thank you again for your helpful comments.

Reviewer 2 Report

The manuscript described the survival outcomes of SIMMILR-2 study. When comparing OLR, LLR and RLR, the OS and RFS were all similar after utilization of the Milan criteria and PSM .The result is interesting, however, I have several concerns listed below. 

1. I suggest to add another table to compare the oncologic factors 

of Open Liver Resection (OLR) vs. minimally invasive(RLR+LLR) 

2. It seemed that the 3 and 5 year OS in RLR group is higher than OLR group (although not statistically significant). Is it possible to include more patients into the two groups?

3.Please rearrange the pictures in this study to make them more concise.

Author Response

RESPONSE TO REVIEWER 2

The manuscript described the survival outcomes of SIMMILR-2 study. When comparing OLR, LLR and RLR, the OS and RFS were all similar after utilization of the Milan criteria and PSM .The result is interesting, however, I have several concerns listed below. 

1. I suggest to add another table to compare the oncologic factors 

of Open Liver Resection (OLR) vs. minimally invasive(RLR+LLR) 

This Table was added and is now labeled TABLE 2.

OLR

n = 164

MILR

n = 206

p-value

Synchronous 

(%)

114

(69.5)

165

(80.1)

0.021

Neoadjuvant Chemotherapy 

(%)

122

(74.4)

148

(71.8)

0.638

Folfox (%)

Average Cycles 

(range)

66(40.2) 

6

(4.5-12)

70 (34.0)

5

(3-12)

0.341

Folfirinox (%)

Average Cycles 

(range)

43 (26.2)

8 

(6-12)

54 (26.2)

6

(3-12)

5-FU alone (%)

Average Cycles

(range)

13 (7.9)

6 

(5-9)

24 (11.7)

4

(3-7)

Neoadjuvant Biologicals 

(%)

63

(38.4)

61

(29.6)

0.078

Bevacizumab (%)

Average Cycles

 (range)

43 (26.2)

6 

(5-12)

46 (22.3)

5 

(4-12)

Cetuximab (%)

Average Cycles 

(range)

20 (12.2)

6 

(5-8)

15 (13.6)

6 

(4-8)

Table 2. Oncologic factors after Propensity Score Matching of all Open Liver Resections (OLR) compared to all Laparoscopic Liver Resections (LLR) and Robotic Liver Resections (RLR) combined, including type of neoadjuvant chemotherapy and biological administered.

And this was added to the RESULTS, to discuss the new TABLE :

When all the OLR patients from the PSM were compared to all of the patients who underwent an minimally invasive liver resection (MILR), LLR and RLR, significantly more synchronous lesions were found in the MILR group when compared to the OLR group, 80% vs. 70%, respectively (p=0.02)(Table 2). Furthermore, fewer biological agents tended to be used MILR group (30%) when compared to the OLR group (38%), but this only approached statistical significance (p= 0.08).

2. It seemed that the 3 and 5 year OS in RLR group is higher than OLR group (although not statistically significant). Is it possible to include more patients into the two groups?

Unfortunately, due to the fact that this is a follow-up study to one that was already published, we are limited to using only those patients. Additionally, he thought about doing Kaplan Meier curves for the combination that you requested in TABLE 2, however, if we did that the results would no longer follow the Propensity Score Matching an dthe results would not be controlled or useful.

3.Please rearrange the pictures in this study to make them more concise.

Figures 1 and 4 were combined into new Figure 1 and labeled A and B, respectively.

Figures 2 and 5 were combined into new Figure 2 and labeled A and B, respectively.

Figures 3 and 6 were combined into new Figure 3 and labeled A and B, respectively.

Tables 1-3 were consolidated into 1 Table at the request of one of the other Reviewers.

The entire Manuscript was extensively modified and edited. Over 4 and 1/2 paragraphs were removed from the Discussion to make it clearer.

A new section called FUTURE DIRECTIONS was added to also help organize the Discussion section and was placed at the end of the DISCUSSION, right before the CONCLUSION.

Future Directions

Although advances in AI will enable enhanced computer vision and more autonomous actions in the operating room, it doesn’t appear that the type of approach used has a significant impact on survival for patients with synchronous CRLMs that adhere to the Milan Criteria. In addition to improvements in chemotherapeutic and biological agents, increased utilization of big data analytic tools in determining which patients are good candidates for surgery, what types of neoadjuvant and adjuvant treatments to administer may help improve OS and RFS even further[4, 12, 55, 70-82]. More complex models may be used in the future to better predict survival. One such example is the contour prognostic model that uses the size of the largest hepatic metastasis, total number of metastases and RAS mutation status[1]. Although not widely used, except in large referral centers, it is hoped that a broader implementation of artificial intelligence (AI) in the future will make the routine use of models such as these more prevalent[2].

Newer collaborative robots that are handheld or do not use a console may also become useful in the surgeon’s armamentarium against CRLM and may actually permit a safer adoption of more automatic and autonomous actions in the operating room[3-6]. Modern day liver surgeons will need expertise in Open, Laparoscopic and Robotic-assisted techniques to offer the best personalized surgical care to their patients with CRLM. An analysis of long-term outcomes may show that the minimally invasive approach has advantages to the surgical management of CRLM once more surgeons are able to receive the proper mentorship and experience in these various techniques[7].

Thank you for your helpful comments.

Reviewer 3 Report

The article “Survival Study: International Multi-Centric Minimally Invasive Liver Resection for Colorectal Liver Metastases (SIMMILR-2)” reports on valuable, long-term outcomes from SIMMILR-1. Specifically, the authors reported similar OS and RFS among OLR, LLR, and RLR. Although the sample sizes in the RLR versus OLR (n=22) and RLR versus LLR (n=21) comparisons were limited after propensity score matching, the careful selection and matching of patients allowed for a fair comparison among operative approaches in as homogenous a cohort as possible. The biggest perceived issue with this manuscript was in presenting a coherent discussion. 

A few questions/comments are shared below:

  1. Overall, the “Discussion” section was scattered and difficult to follow. For example, lines 207-212 mention the importance of following patients for 10-years, even though this study follows the patients for 5 years. In the following paragraph (lines 220-222), the authors discuss that patients who undergo LLR are more likely to undergo repeat hepatectomy for recurrence - this is not an outcome that was explored in this paper, and thus it seems extraneous to discuss. In another example, in the few paragraphs in lines 266 onward, the authors go into detail about the impact of margin status on survival, CALI, etc. Again, while some of these concepts are worth mentioning briefly if directly relevant to the study, the level of current detail was unnecessary and made this section very lengthy and difficult to read.  
  2. The introduction could also use some minor changes for improved readability/organization; for example, it may be worthwhile having the first paragraph include more general, background information, followed by a paragraph on SIMMILR-1 with pertinent results, followed by the current study/its importance. More specifically, the sentence in lines 46-51 seems better served in the discussion/conclusion (i.e. future direction) and not in the introduction. 
  3. Errors in grammar/syntax throughout the manuscript should be addressed. 
  4. Recommend including more information on methodology. The authors often refer back to SIMMILR-1 which made it difficult for the reader to flip back and forth between papers; if too much information to include in the body of the manuscript, please include as appendices. 

Author Response

COMMENTS TO REVIEWER 3

The article “Survival Study: International Multi-Centric Minimally Invasive Liver Resection for Colorectal Liver Metastases (SIMMILR-2)” reports on valuable, long-term outcomes from SIMMILR-1. Specifically, the authors reported similar OS and RFS among OLR, LLR, and RLR. Although the sample sizes in the RLR versus OLR (n=22) and RLR versus LLR (n=21) comparisons were limited after propensity score matching, the careful selection and matching of patients allowed for a fair comparison among operative approaches in as homogenous a cohort as possible. The biggest perceived issue with this manuscript was in presenting a coherent discussion. 

A few questions/comments are shared below:

  1. Overall, the “Discussion” section was scattered and difficult to follow. For example, lines 207-212 mention the importance of following patients for 10-years, even though this study follows the patients for 5 years. In the following paragraph (lines 220-222), the authors discuss that patients who undergo LLR are more likely to undergo repeat hepatectomy for recurrence - this is not an outcome that was explored in this paper, and thus it seems extraneous to discuss. In another example, in the few paragraphs in lines 266 onward, the authors go into detail about the impact of margin status on survival, CALI, etc. Again, while some of these concepts are worth mentioning briefly if directly relevant to the study, the level of current detail was unnecessary and made this section very lengthy and difficult to read.  

The DISCUSSION was drastically shortened with almost 5 entire paragraphs being removed and extensive modification of this section. A new section entitled FUTURE DIRECTIONS was also placed at the end of the DISCUSSION to better organize our thoughts on the potential role of Artificial Intelligence and future robotics in the management of patients with CRLM.

Future Directions

Although advances in AI will enable enhanced computer vision and more autonomous actions in the operating room, it doesn’t appear that the type of approach used has a significant impact on survival for patients with synchronous CRLMs that adhere to the Milan Criteria. In addition to improvements in chemotherapeutic and biological agents, increased utilization of big data analytic tools in determining which patients are good candidates for surgery, what types of neoadjuvant and adjuvant treatments to administer may help improve OS and RFS even further[4, 12, 55, 70-82]. More complex models may be used in the future to better predict survival. One such example is the contour prognostic model that uses the size of the largest hepatic metastasis, total number of metastases and RAS mutation status[1]. Although not widely used, except in large referral centers, it is hoped that a broader implementation of artificial intelligence (AI) in the future will make the routine use of models such as these more prevalent[2].

Newer collaborative robots that are handheld or do not use a console may also become useful in the surgeon’s armamentarium against CRLM and may actually permit a safer adoption of more automatic and autonomous actions in the operating room[3-6]. Modern day liver surgeons will need expertise in Open, Laparoscopic and Robotic-assisted techniques to offer the best personalized surgical care to their patients with CRLM. An analysis of long-term outcomes may show that the minimally invasive approach has advantages to the surgical management of CRLM once more surgeons are able to receive the proper mentorship and experience in these various techniques[7].

  1. The introduction could also use some minor changes for improved readability/organization; for example, it may be worthwhile having the first paragraph include more general, background information, followed by a paragraph on SIMMILR-1 with pertinent results, followed by the current study/its importance. More specifically, the sentence in lines 46-51 seems better served in the discussion/conclusion (i.e. future direction) and not in the introduction. 

The INTRODUCTION was heavily edited and streamlined based on your insightful recommendations. As mentioned above, we also took your advice and moved part of the INTRODUCTION to the section entitled FUTURE DIRECTIONS. 

The INTRODUCTION now reads :

The Study: of International Multi-centered Minimally Invasive Liver Resection (SIMMILR) was a study focused on ColoRectal Liver Metastases (CRLM) from 6 international centers where after Propensity Score Matching (PSM), data from only 5 centers remained[8]. In an effort to get as fair a comparison as possible, only CRLM that adhered to the Milan Criteria, a classification system traditionally reserved for hepatocellular carcinoma[9], but increasingly used for CRLM, were studied [1]. Potential bias between liver resections performed through an open approach and minimally invasive techniques was reduced by controlling for preoperative factors. The initial study (SIMMILR-CRLM), from here on designated as SIMMILR-1, focused on an analysis of short-term outcomes, and compared open liver resection (OLR), laparoscopic (LLR) and robotic-assisted liver resection (RLR) to eachother[8]. After PSM in SIMMILR-1, blood loss, hospitalization and morbidity rates were statistically significantly decreased in the LLR group when compared to the OLR group. RLR had significantly less estimated blood loss when compared to both OLR and LLR.

This study, SIMMILR-2, aims to assess whether any difference can be found between a comparison of these 3 surgical approaches in terms of long-term outcomes, namely overall survival (OS) and recurrence free survival (RFS). Data exists comparing OLR and LLR, but is limited to short-term outcomes with oncologic data limited to showing equivalency of resection margin status[10]; similarly, short-term outcomes[11, 12]and margin status has been compared between OLR and RLR for CRLM with similar results seen in both groups[13, 14]. Due to the great disparity in how studies and data are reported, with synchronous and metachronous lesions often being reported together, we only reported data from patients who presented with synchronous lesions, and we controlled for neoadjuvant chemotherapeutic and biological treatments received.

  1. Errors in grammar/syntax throughout the manuscript should be addressed. 

The manuscript was HEAVILY edited by a native english speaker, spell check was utilized several times and extra spaces were sought out and eliminated.

  1. Recommend including more information on methodology. The authors often refer back to SIMMILR-1 which made it difficult for the reader to flip back and forth between papers; if too much information to include in the body of the manuscript, please include as appendices. 

The relevant findings from SIMMILR-1 were added to the INTRODUCTION. The METHODS section was also edited to better clarify the relevant points from SIMMILR-1. 

Upon request by one of the other Reviewers, Tables 1-3 were combined into a single Table now labeled TABLE 1. A new requested Table was also added and was labeled TABLE 2. It combines all of the patients after Propensity Score Matching (PSM) who underwent Open Liver Resection and compares them with all of the patients who underwent either Laparoscopic or Robotic Liver resection. Survival Curves of this Table were not included because these 2 comparison groups were no longer controlled by PSM.

Lastly, another Reviewer requested that I combine the Survival Curves into fewer FIGURES.

Figures 1 and 4 were combined into new Figure 1 and labeled A and B, respectively.

Figures 2 and 5 were combined into new Figure 2 and labeled A and B, respectively.

Figures 3 and 6 were combined into new Figure 3 and labeled A and B, respectively.Bibliography
